# The microRNA cluster miR-183/96/182 contributes to long-term memory in a protein phosphatase 1-dependent manner

Bisrat T. Woldemichael[1,*,†], Ali Jawaid[1,*], Eloïse A. Kremer[1], Niharika Gaur[1], Jacek Krol[2], Antonin Marchais[3] & Isabelle M. Mansuy[1]

Memory formation is a complex cognitive function regulated by coordinated synaptic and nuclear processes in neurons. In mammals, it is controlled by multiple molecular activators and suppressors, including the key signalling regulator, protein phosphatase 1 (PP1). Here, we show that memory control by PP1 involves the miR-183/96/182 cluster and its selective regulation during memory formation. Inhibiting nuclear PP1 in the mouse brain, or training on an object recognition task similarly increases miR-183/96/182 expression in the hippocampus. Mimicking this increase by miR-183/96/182 overexpression enhances object memory, while knocking-down endogenous miR-183/96/182 impairs it. This effect involves the modulation of several plasticity-related genes, with *HDAC9* identified as an important functional target. Further, PP1 controls miR-183/96/182 in a transcription-independent manner through the processing of their precursors. These findings provide novel evidence for a role of miRNAs in memory formation and suggest the implication of PP1 in miRNAs processing in the adult brain.

[1] Laboratory of Neuroepigenetics, University of Zurich/Swiss Federal Institute of Technology, Brain Research Institute, Neuroscience Center Zürich, Zurich CH-8057, Switzerland. [2] Friedrich Miescher Institute for Biomedical Research, Basel CH-4048, Switzerland. [3] Institute of Agricultural Sciences, Swiss Federal Institute of Technology, Zurich CH-8092, Switzerland. * These authors contributed equally to this work. † Present address: Department of Psychiatry, Icahn School of Medicine at Mount Sinai, New York, New York 10029, USA. Correspondence and requests for materials should be addressed to I.M.M. (email: mansuy@hifo.uzh.ch).

The formation of long-term memory depends on synaptic plasticity and activity-dependent structural and functional changes in neuronal circuits. It is sustained by cascades of tightly orchestrated signalling molecules that positively or negatively regulate synaptic efficacy for the control of memory formation[1,2]. One of the ultimate functions of these signalling cascades is the regulation of gene expression and the synthesis of new proteins necessary for the formation and the storage of long-term memory[3,4]. In these cascades, protein kinases such as $Ca^{2+}$/calmodulin-dependent protein kinase II (CaMKII) and the cAMP-dependent protein kinase A (PKA) have a permissive role while protein phosphatases such as $Ca^{2+}$/calmodulin-dependent protein phosphatase calcineurin and protein phosphatase 1 (PP1) act as memory suppressors[5,6].

PP1 is a ubiquitous phosphatase in the brain well positioned to orchestrate molecular processes resulting from neuronal activity, and is implicated in many brain functions. Our previous work has demonstrated that PP1 can regulate the expression of genes important for memory formation by influencing the epigenetic state of these genes, in particular, through post translational modifications of histone proteins[7–10]. PP1 is also an important regulator of gene transcription and ribonucleic acid (RNA) processing[11–13].

In recent years, microRNAs (miRNAs) have emerged as important regulators of gene expression in many biological systems. Most miRNAs identified so far are expressed in the brain and have distinct expression patterns at different developmental stages, and in different brain regions and cell types[14–16]. In addition, many miRNAs, along with components of the miRNA biogenesis and silencing machinery are enriched at synaptic sites[17]. So far, several miRNAs have been implicated in neuronal functions including learning and memory formation[18,19]. Yet, their mode of regulation in the context of cognitive functions remains poorly understood.

Here we show that a cluster of miRNAs comprising miR-183/96/182 is differentially regulated upon learning and is modulated by PP1. We provide evidence that the production of miR-183/96/182 precursors is favored by the inhibition of PP1 in a transcription-independent manner and that overexpression of the cluster in the hippocampus enhances memory in adult mice while its knockdown impairs memory. These effects are proposed to be mediated by regulation of miR-183/96/182 biogenesis and suppression of target genes such as HDAC9.

## Results

**miR-183/96/182 is upregulated by PP1 inhibition or learning.** To examine the role of PP1 in the regulation of miRNAs involved in memory formation, we took advantage of a transgenic mouse line in which the activity of PP1 can be inhibited inducibly in adult forebrain neurons by expression of a fragment of a nuclear inhibitor of PP1, NIPP1 (NIPP1*) (ref. 9). Previous work established that in these mice, inhibition of nuclear PP1 improves hippocampus-dependent forms of memory and causes widespread epigenetic and transcriptional changes of several genes[8–10]. Based on these findings and considering that PP1 can act as a transcriptional and post-transcriptional gene regulator, we postulated that inhibition of nuclear PP1 may alter the expression of miRNAs important for memory formation.

To test this hypothesis, NIPP1* animals and control littermates were trained on a novel object recognition (NOR) task (Supplementary Fig. 1). We chose NOR because it is a paradigm based on the natural attraction of rodents for novelty that involves hippocampus-dependent functions[20,21], and can be used to test short- and long-term memory. Performance on the NOR task is also known to be modulated by PP1 (refs 7,9). Following

NOR training, miRNAs were examined in the hippocampus by next-generation deep sequencing. In total, over 84 million reads were sequenced, a large proportion of which (83%) corresponded to a size of 19–26 nt with over 92% mapping to known mouse miRNAs (Supplementary Fig. 2a,b). The expression level of the identified miRNAs varied greatly, with some miRNAs being highly abundant, and others moderately or lowly abundant (Supplementary Fig. 2c). Overall, the level of most miRNAs was consistent across samples in NIPP1* transgenic and control littermates whether trained or not (Supplementary Fig. 2d), suggesting no gross alteration of miRNAs expression by PP1 inhibition or NOR training. There was also no global change in the expression of major components of the miRNA biogenesis machinery by PP1 inhibition (Supplementary Fig. 3).

Differential expression analyses revealed that distinct sets of miRNAs are upregulated or downregulated in NIPP1* mice compared with control littermates and in NOR-trained mice compared with non-trained animals. Notably, a subset of miRNAs was similarly altered by PP1 inhibition and training (Fig. 1a,b, Supplementary Figs 4 and 5). A closer look at these miRNAs revealed that the miR-183/96/182 cluster is upregulated in the hippocampus in both, NIPP1* transgenic mice and NOR-trained controls. Quantitative polymerase chain reaction (qPCR) confirmed a consistent increase (about 50%) in miR-183 and miR-182, while miR-96 was expressed at low level (Fig. 1c,d). miR-183/96/182 have been implicated in neuronal activity and plasticity as well as in amygdala-dependent fear memory[22–24], and their predicted targets are involved in plasticity and neuronal signalling pathways.

To confirm the link between PP1 inhibition and the upregulation of miR-183/96/182 cluster, PP1 was knocked down in N2A cells using a pool of siRNAs targeting the 3′UTR of PP1γ, an isoform predominantly linked to nuclear functions (Supplementary Fig. 6a). PP1γ knockdown increased the level of miR-183 and miR-182 (Supplementary Fig. 6b), confirming that nuclear PP1 is implicated in the synthesis of these miRNAs.

**PP1 inhibition affects miR-183/96/182 biogenesis.** We next examined the potential link between miR-183/96/182 cluster and PP1. miRNAs are produced through a succession of biogenesis steps involving the transcription of primary miRNAs (pri-miRs) and their processing into precursor miRNAs (pre-miRs) in the nucleus then to mature miRNAs in the cytoplasm[25]. To determine whether miR-183/96/182 biogenesis is modulated by PP1, we measured the level of pre-miR-183/96/182 in NIPP1* animals. In the hippocampus, pre-miR-183 and pre-miR-182 were upregulated in the nuclear fraction and downregulated in the cytoplasmic fraction (Fig. 2a,b). Similarly, neuronal activity induced by KCl treatment led to a rapid upregulation of pri-miR-183/96/182 and corresponding pre-miR transcripts in N2A cells (Supplementary Fig. 6c). Combining PP1γ knockdown and KCl stimulation caused a further increase in nuclear pre-miRNAs, which was reversed by overexpression of a PP1γ construct carrying a siRNA-resistant open reading frame (Fig. 2c). Further, PP1γ knockdown reduced the level of KCl-induced pri-miR-183/96/182 and cytoplasmic pre-miR-183/96/182 but had no effect on a control miRNA (Fig. 2d, Supplementary Fig. 6d,e), suggesting a selective effect. We next examined whether this action of PP1 requires gene transcription using the transcription inhibitor actinomycin D (ActD) (Supplementary Fig. 7). While ActD treatment significantly reduced pri-miRNA transcript (Fig. 2d), it had minimal effect on the up-regulation of pre-miRs induced by PP1γ knockdown (Fig. 2e), suggesting that PP1γ inhibition likely acts downstream of RNA Pol II-dependent transcription to regulate miR-183/96/182 level.

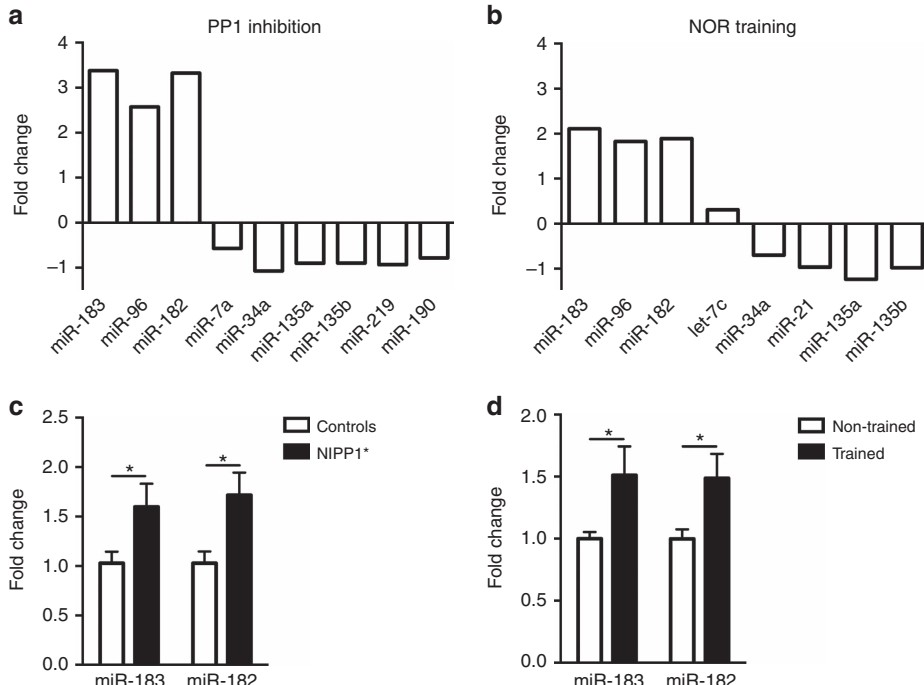

**Figure 1 | PP1 inhibition and NOR training induce differential expression of several miRNAs in the mouse hippocampus.** NIPP1* and control mice were trained on an NOR task (trained) or exposed to the training chamber alone (non-trained). Deep sequencing was conducted on pooled hippocampal samples ($n = 6$ in each group) collected 30 min after the end of training. Several miRNAs are differentially expressed (adjusted $P < 0.05$) in non-trained NIPP1* mice compared with non-trained controls (**a**, expressed as log 2 of fold change), and in trained controls compared with non-trained controls (**b**, expressed as log 2 of fold change). Some of the miRNAs identified by deep sequencing were similarly upregulated in an independent set of experiments in both non-trained NIPP1* mice (**c**, miR-183: controls, $n = 11$; NIPP1*, $n = 11$; t20 = 2.19, *$P < 0.05$; miR-182: controls, $n = 11$; NIPP1*, $n = 11$; t20 = 2.68, *$P < 0.05$); and control mice trained on NOR (**d**; miR-183: non-trained, $n = 12$; trained, $n = 13$; t23 = 2.07, *$P \leqslant 0.05$; miR-182: non-trained, $n = 13$; trained, $n = 13$; t24 = 2.32, *$P < 0.05$). Bar graphs represent mean ± s.e.m.

To further explore the mechanisms by which PP1 inhibition influences the level of the cluster's pre-miRs, we conducted pri-miRNA processing assays. In these assays, cropping of the stem-loop hairpin sequence from artificially introduced pri-miRNA transcript, indicating pri-miRNA processing, can be monitored by reduced firefly luciferase signal[26] (Fig. 3a). We observed that processing of both pri-miR-183 and pri-miR-182, but not a control pri-miR-10b, was significantly increased upon inhibition of nuclear PP1 by *NIPP1* overexpression (Fig. 3b–d). Together, these results suggest that inhibition of PP1γ facilitates the increase in miR-183/96/182 level triggered by neuronal activity by favoring pre-miRNA production at the microprocessor level.

**PP1 inhibition replenishes the existing pool of miRNAs.** Most neuronal miRNAs have a fairly rapid turnover following cellular activity, where changes in pri-miRNA transcription and processing precede changes in the level of mature miRNAs[23]. To examine whether this process is affected by PP1, we measured the level of miR-183/96/182 transcripts at different time points, with and without transcriptional inhibition. We observed that KCl stimulation decreases the level of mature miR-183/96/182 after 30 min but significantly increases it after 4 h (Fig. 4a,b). The early decrease was not affected by ActD treatment, but the increase at 4 h was reversed and even inversed (leading to a decrease) by ActD (Fig. 4c,d), suggesting an initial usage and depletion of miRNAs induced by activity, followed by replenishment through increased biogenesis. Blockade of this replenishment by ActD results in continued miRNAs depletion. This blockade could be partially rescued by *PP1γ* inhibition (Fig. 4e), confirming that PP1 acts downstream of gene transcription.

**MiR-183/96/182 modulation affects long-term memory.** Since the miR-183/96/182 cluster is upregulated in the adult hippocampus following NOR training, we next examined whether inducing its expression in the hippocampus at the time of learning affects object memory. We overexpressed miR-183/96/182 in hippocampus area CA1 in adult mice *in vivo* using a self-complementary adeno-associated virus (scAAV) vector expressing truncated pri-miR-183/96/182 fused with *GFP*. Virus transduction and miRNA overexpression were confirmed by immunohistochemistry and qPCR (Supplementary Fig. 9). For NOR training (acquisition), the animals were exposed to three unfamiliar objects for five sessions of 5 min spaced by 5 min intervals, a protocol that induces robust long-term memory[7] (Supplementary Fig. 10a). miR-183/96/182 overexpressing mice and controls similarly explored the objects during acquisition (Supplementary Fig. 10b). Both groups had comparable long-term object memory when tested 24 h after training (Supplementary Fig. 10c,d). Overall locomotor activity was similar in mice overexpressing miR-183/96/182 and controls (Supplementary Fig. 10e,f).

Previous studies on NOR and other memory paradigms have demonstrated that the duration and spacing of training sessions determine memory strength. For most paradigms, repeated and spaced training results in stronger memory than massed training[7,27–29]. Because our NOR training protocol was repeated and spaced, it elicited strong memory that may have masked the effect of miR-183/96/182 overexpression. Thus, we repeated the experiment using a weaker protocol based on a single 10 min training session followed by two test sessions 24 h apart (weak protocol, Fig. 5a). Similar to the strong protocol, training with the weak protocol increases the expression of precursor and mature forms of miR-183/96/182 cluster (Supplementary Figs 8 and 13). Comparison

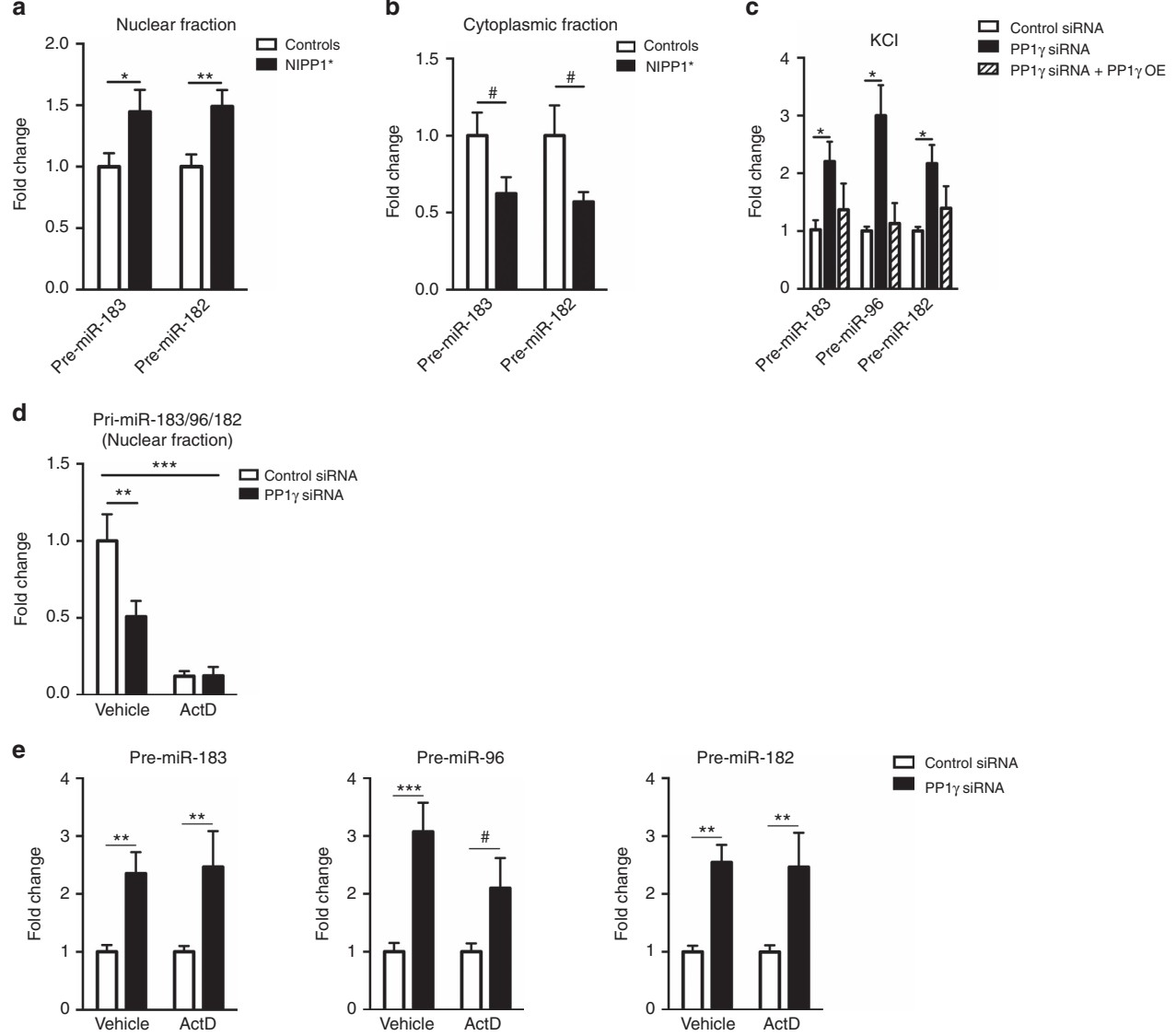

**Figure 2 | PP1 inhibition up-regulates pre-miR183/96/182 expression. (a)** Pre-miR-183/96/182 expression in the nuclear fraction of the hippocampus from NIPP1* and control mice (pre-miR-183: controls, $n = 11$; NIPP1*, $n = 11$; $t20 = 2.12$, *$P < 0.05$; pre-miR-182: controls, $n = 11$; NIPP1*, $n = 11$; $t20 = 2.92$, **$P < 0.01$). **(b)** Cytoplasmic pre-miR-183/96/182 level in the hippocampus of NIPP1* mice and control littermates (pre-miR-183: $t10 = 2.05$, #$P < 0.1$; pre-miR-182: $t10 = 2.09$, #$P < 0.1$; controls, $n = 6$; NIPP1, $n = 6$). **(c)** PP1γ knockdown combined with KCl stimulation (1 h) causes upregulation of pre-miR-183, pre-miR-96, and pre-miR-182, which is reversed by PP1γ overexpression (pre-miR-183: one-way ANOVA, F2,6 = 3.12, $P = 0.12$; $t$-test: t(4) = 3.09, *$P < 0.05$; pre-miR-96: one-way ANOVA, F2,6 = 9.19, $P = 0.01$; $t$-test: t(4) = 3.75, *$P < 0.05$; pre-miR-182: one-way ANOVA, F2,6 = 4.13, $P = 0.07$; $t$-test: t(4) = 3.52, *$P < 0.05$). **(d)** Nuclear pri-miRNA regulation by PP1γ knockdown and the effect of ActD treatment in N2A cells after 1 h of KCl stimulation (two-way ANOVA, ActD: F1,8 = 35.62, ***$P < 0.001$, PP1γ: F1,8 = 5.37, $P < 0.05$, *post-hoc*: vehicle **$P ≤ 0.01$). **(e)** ActD treatment of N2A cells does not fully abolish pre-miRNA upregulation induced by PP1γ knockdown; left panel: pre-miR-183, two-way ANOVA: PP1γ-F(1,27) = 17.6, $P = 0.0003$ (*post-hoc*: vehicle t27 = 3.13, **$P < 0.01$; ActD t27 = 2.85, **$P < 0.01$); middle panel: pre-miR-96, two-way ANOVA: PP1γ-F(1,27) = 16.1, $P < 0.001$ (*post-hoc*, vehicle t27 = 4.05, ***$P < 0.001$; ActD t27 = 1.8, #$P < 0.1$); right panel: pre-miR-182, two-way ANOVA: PP1γ-F(1,28) = 21.3, $P < 0.0001$ (*post-hoc*, vehicle t28 = 3.45, **$P < 0.01$; ActD t28 = 3.11, **$P < 0.01$). Bar graphs represent mean ± s.e.m.

of miR-183/96/182 overexpressing mice and controls with this protocol showed no difference between the groups in overall locomotor activity in an open field test or during training in the NOR task (Supplementary Fig. 11a–c). During test 1, both groups had a comparable low level of memory, which increased during test 2 (Supplementary Fig. 11d,e), consistent with the notion that retrieval helps update and strengthen memory[30,31]. Importantly, during the second test, miR-183/96/182 overexpressing mice had significantly better memory than control mice (Fig. 5b).

To confirm the implication of miR-183/96/182 in long-term memory, we also expressed a sponge construct that competitively

inhibits the miRNA cluster in the mouse hippocampus (Supplementary Fig. 12a,b). While sponge expression did not affect overall locomotion or object exploration (Supplementary Fig. 12c,d), it significantly impaired long-term memory 24 h after training (Fig. 5c, Supplementary Fig. 12e). Taken together, these results provide evidence for a permissive role of the miR-183/96/182 cluster in the hippocampus in long-term object memory.

**miR-183/96/182 regulates plasticity-related genes.** Many of the predicted targets of the miR-183/96/182 cluster are involved in biological pathways relevant for neuronal signalling and

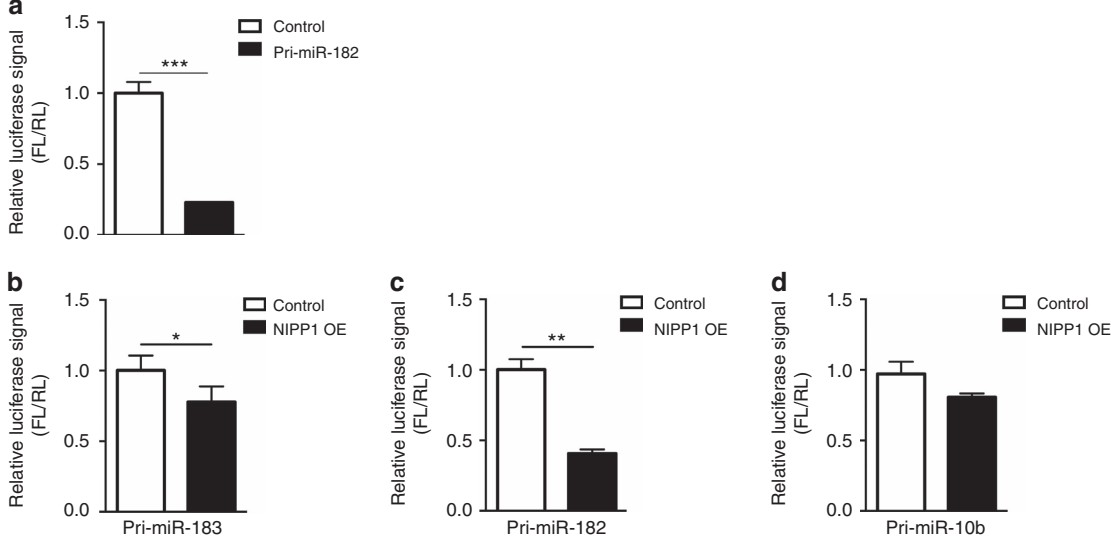

**Figure 3 | Analyses of pri-miRNA processing.** (**a**) Relative signal of firefly luciferase (FL) to renilla luciferase (RL) is reduced in pmirGLO_pri-miR-182 transfected cells (Pri-miR-182) compared with control cells transfected with an empty vector (Control) (t4 = 9.8, \*\*\*$P<0.001$). (**b–d**) Overexpression of a nuclear inhibitor of PP1 (NIPP1 OE) reduces FL/RL signal in cells expressing pri-miR-183 (**b**, t5 = 2.71, \*$P<0.05$) and pri-miR-182 (**c**, t4 = 7.45, \*\*$P<0.01$), but not pri-miR-10b (**d**, t4 = 1.83, $P=0.14$), fused to a luciferase reporter. Bar graphs represent mean ± s.e.m.

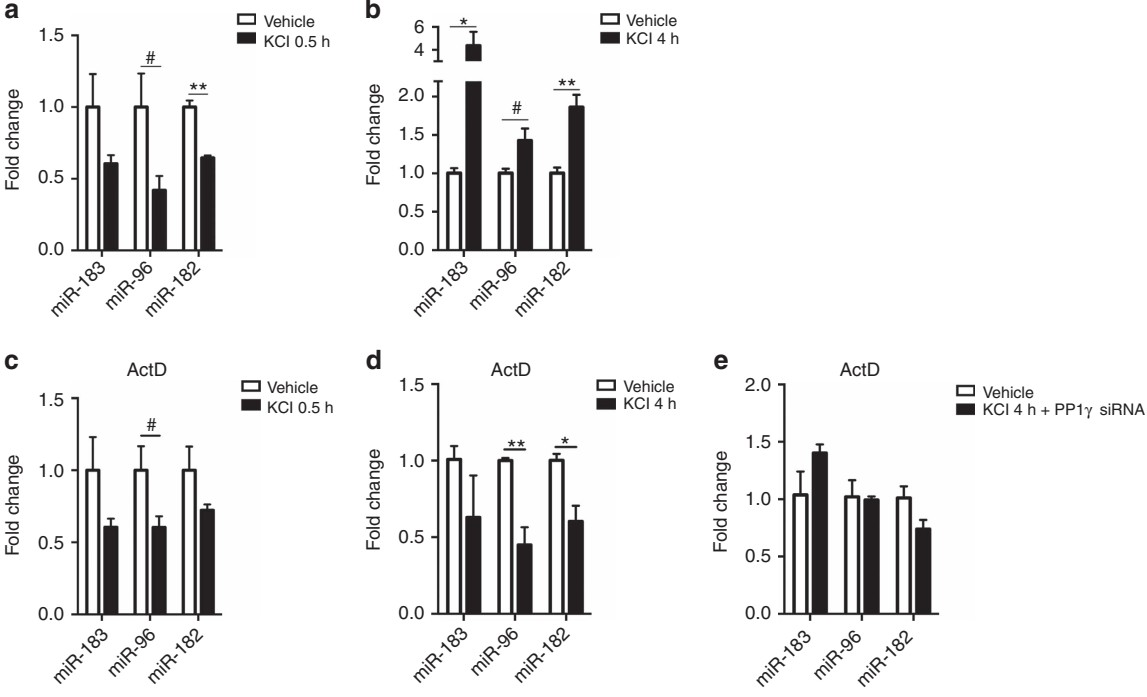

**Figure 4 | PP1 inhibition facilitates an increase in mature miRNAs following neuronal stimulation.** (**a**) Mature miRNAs level is lower 0.5 h after KCl treatment (miR-183: t4 = 1.66, $P=0.17$; miR-96: t4 = 2.28, #$P<0.1$; miR-182: t4 = 7.2, \*\*$P<0.01$). (**b**) Mature miRNAs level is increased 4 h after KCl treatment (miR-183: t4 = 2.81, \*$P<0.05$; miR-96, t4 = 2.55, #$P<0.1$; miR-182, t4 = 4.86, \*\*$P<0.01$). (**c**) Effect of ActD treatment and 0.5 h of KCl stimulation on mature miRNA level (miR-183, t4 = 1.66, $P=0.17$; miR-96, t4 = 2.17, #$P<0.1$; miR-182, t4 = 1.66, $P=0.18$). (**d**) ActD treatment reduces the level of mature miRNAs 4 h after KCl stimulation (miR-183, t4 = 1.32, $P=0.26$; miR-96, t4 = 4.68, \*\*$P<0.01$; miR-182, t4 = 3.59, \*$P<0.05$). This effect of ActD is prevented by PP1γ knockdown (**e**: miR-183, t4 = 1.68, $P=0.17$; mir-96, t4 = 0.20, $P=0.85$; miR-182, t4 = 2.09, $P=0.10$). Bar graphs represent mean ± s.e.m.

plasticity, and epigenetic regulation (Supplementary Fig. 17). To validate some of these targets, we measured their level of expression in the hippocampus of mice overexpressing miR-183/96/182. Several genes including ion channels, receptors, a kinase, a phosphatase and a histone deacetylase (HDAC) were significantly downregulated (Fig. 6a). We focused on one of these genes, *HDAC9*, which codes for a member of class II HDACs

known to be an epigenetic regulator modulated by neuronal activity, and is involved in the control of plasticity-related genes[32–35]. Histone acetylation and the enzymes that modulate acetylation such as HDACs, play a crucial role in the formation and storage of long-term memory[36] and HDAC inhibitors are potent drugs for correcting cognitive deficits[37]. We therefore explored the link between the miR-183/96/182 cluster and

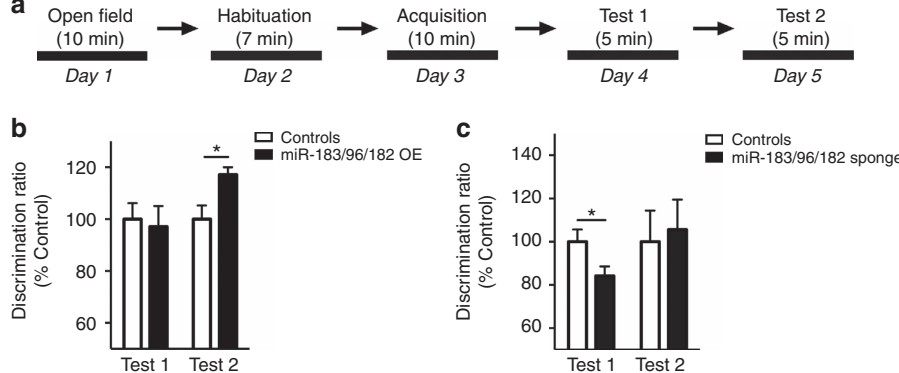

**Figure 5 | miR-183/96/182 modulation in the hippocampus affects long-term object memory.** (**a**) Experimental setup. (**b**) Object discrimination ratio (expressed as per cent of controls) in miR-183/96/182 overexpressing mice and controls during memory test 24 h (Test 1, t32 = 0.298, P = 0.77) and 48 h (Test 2, t32 = 2.65, *P < 0.05) after acquisition; controls, n = 19; miR-183/96/182, n = 15. (**c**) Object discrimination ratio (expressed as per cent of controls) in miR-183/96/182 sponge expressing mice and controls during memory test at 24 h (Test 1, t18 = 2.18, *P < 0.05) and 48 h (Test 2, t16 = 1.079, P = 0.297); controls, n = 9–10, miR-183/96/182 sponge, n = 10–11. Bar graphs represent mean ± s.e.m.

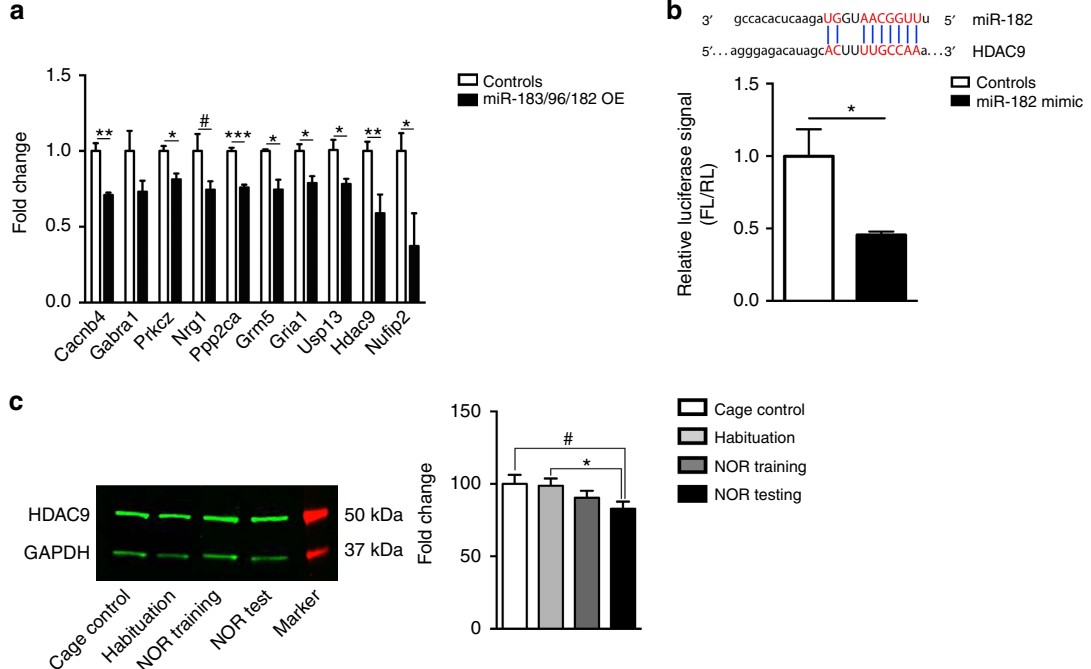

**Figure 6 | miR-183/96/182 cluster decreases the expression of genes involved in plasticity and altered by NOR training, including HDAC9.** (**a**) Expression level of target genes in the hippocampus of miR-183/96/182 overexpressing mice: Cacnb4 (t6 = 5.42, **P < 0.01), Gabra1 (t6 = 1.768, P = 0.127), Prkcz (t6 = 3.675, *P < 0.05), Nrg1 (t5 = 2.212, #P < 0.1), Ppp2ca (t6 = 8.159, ***P < 0.001), Grm5 (t5 = 3.246, *P < 0.05), Gria1 (t6 = 3.261, *P = 0.05), Usp13 (t6 = 2.98, *P = 0.05), Hdac9 (t10 = 3.25, **P < 0.01) and Nufip2 (t10 = 2.74, *P < 0.05); controls, n = 3–7; miR-183/96/182, n = 4–5. (**b**) Top panel: Predicted target site of miR-182 seed sequence on HDAC9 3′ UTR. The seed sequence and its corresponding target sequence are highlighted in red (adapted from www.microrna.org); bottom panel: Relative luciferase activity (firefly luciferase (FL) to renilla luciferase (RL)) of HDAC9 3′UTR containing construct measured in N2A cells in the presence or absence of miR-182 mimic (t7 = 2.88, *P < 0.05). (**c**) Representative blot (left panel) and quantification (right panel) of HDAC9 protein in mouse hippocampal extracts after NOR training (one-way ANOVA: F3,28 = 2.06, P = 0.13; t-test: cage control versus NOR testing t(14) = 2.05, #P < 0.1, habituation versus NOR testing t(13) = 2.24, *P < 0.05; cage control, n = 9, habituation, n = 8, NOR training, n = 8, NOR testing, n = 7). Bar graphs represent mean ± s.e.m.

HDAC9. We examined if HDAC9 is a direct target of miR-183/96/182 using a luciferase-based expression system containing a predicted miR-182 binding site of HDAC9 3′UTR. Upon miR-182 transfection, we observed destabilization of the construct, indicating targeting of the predicted HDAC9 target site by miR-182 (Fig. 6b). Further, the level of HDAC9 was reduced in the hippocampus of mice subjected to NOR training and testing compared with controls (Fig. 6c). To further evaluate the importance of HDAC9 targeting by miR-183/96/182, we interfered with miR-182/HDAC9 interaction in vivo by injecting locked nucleic acid (LNA) modified target site blockers (TSB) (Supplementary Fig. 14). Interfering with miR-182 targeting of HDAC9 in mice overexpressing miR-183/96/182 cluster significantly reduced object exploration during training and testing under the weak NOR protocol, without affecting overall locomotion or novel object discrimination (Fig. 7a–e). Together, these results identify HDAC9 as one of the mediators of miR-183/96/182 on cognitive processes.

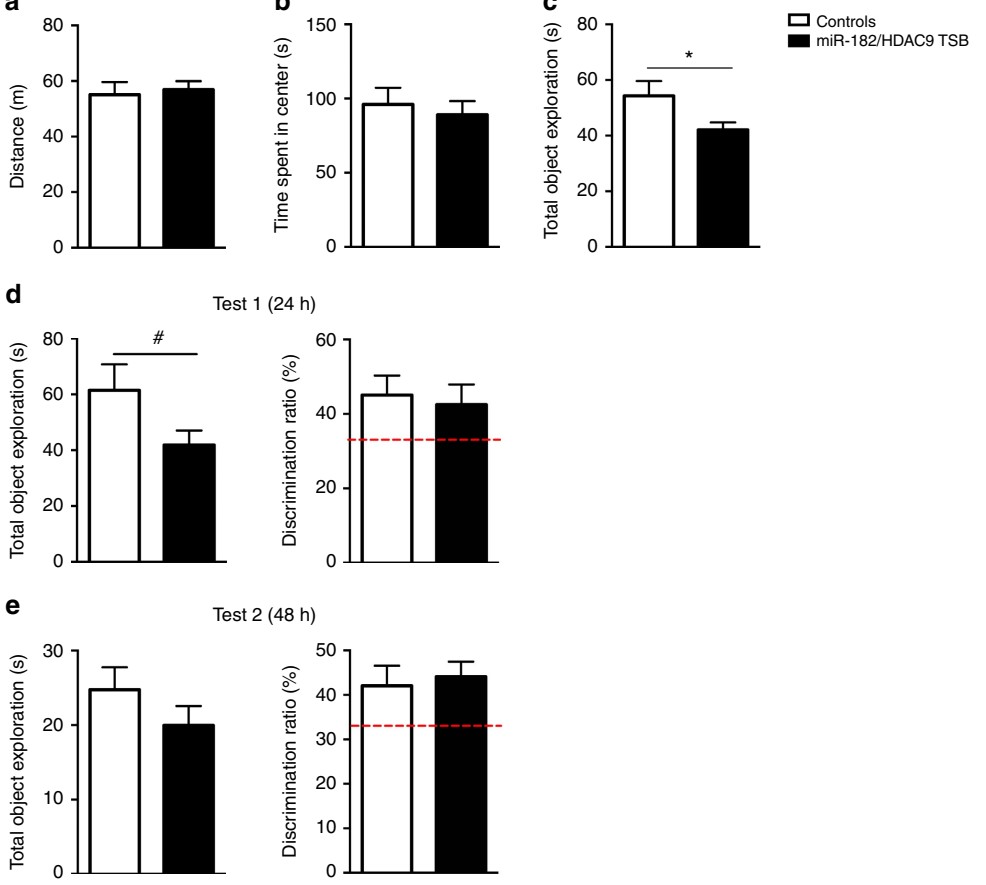

**Figure 7 | Effect of miR-182/HDAC9 TSB on NOR performance in mice overexpressing miR-183/96/182.** (a–c) Behavioural measures during acquisition: total distance covered (**a**, t15 = 0.333, P = 0.74); total time spent in center (**b**, t15 = 0.481, P = 0.64); and total time spent exploring the objects (**c**, t15 = 2.12, *P < 0.05). (**d**) Left panel: total time spent exploring objects during Test 1 (t15 = 1.90, #P < 0.1). Right panel: discrimination ratio of novel versus familiar objects during Test 1 (t15 = 0.343, P = 0.74). (**e**) Left panel: total time spent exploring objects during Test 2 (t15 = 1.22, P = 0.24). Right panel: discrimination ratio of novel versus familiar objects during Test 2 (t15 = 1.04, P = 0.32). Controls, n = 9; miR-182/HDAC9 TSB, n = 8. Bar graphs represent mean ± s.e.m; red broken line indicates chance level of performance.

## Discussion

Our results identify the memory suppressor PP1 as a novel regulator of the biogenesis of miRNAs during memory formation. They show that PP1 inhibition increases the level of the miR-183/96/182 cluster, and identify this cluster as an important modulator of memory formation. *PP1* inhibition acts by enhancing the production of pre-miRNAs in the nucleus. This occurs in a background of mature miR-183/96/182 consumption upon neuronal stimulation and continued replenishment by increased transcription. These findings suggest that *PP1* inhibition facilitates nuclear miRNA processing during neuronal activity, possibly by influencing the microprocessor complex.

Several studies have reported that protein phosphorylation is necessary for pri-miRNA processing, mainly by protein kinase-mediated regulation of the stability, interaction and nuclear localization of components of the microprocessor complex[38–41]. Our study significantly extends these findings by newly showing that PP1 is involved in this processing. The precise modes of action of PP1 remain unknown but PP1 may modulate the processing of specific miRNAs by interacting with RNA binding proteins that contain a PP1 recognition motif[42]. It could also act on splicing since it can interact with components of the spliceosome machinery, known to crosstalk with nuclear miRNA processing[43–46]. Binding of proteins such as Tra2β1, SF2A, Srp30c and ASF to PP1 through conserved RNA recognizing domains is essential for correct splice site selection[42,47], and by

extension may also affect miRNA processing. Interestingly, a recent study demonstrated a developmentally timed processing of pri-miR-183/96/182 involved in neuronal organization. This is mediated by its interaction with Ddx3x, a component of the microprocessor complex[48]. Our results corroborate the highly regulated processing that this cluster undergoes in neurons in response to external signals.

The present results further show that both overexpression and knockdown of the miR-183/96/182 cluster in the mouse hippocampus alter performance in novel object discrimination. While miR-183/96/182 overexpression improves object memory, miR-183/96/182 knockdown impairs memory. Even if the effect size is modest (15–20% increase or decrease in performance), such bidirectional effect is a strong indication that miR-183/96/182 cluster is directly implicated in memory formation with no obvious functional redundancy or compensation. Notably, this effect is observed following training with a weak (single session) protocol but not with a strong (multiple spaced sessions) protocol. This is likely attributable to a 'spacing effect' of training where repeated sessions separated over a period of time lead to stronger activation of molecular pathways needed for memory formation, including plasticity-related proteins[28]. Further, miR-183/96/182 is only one among many other targets of PP1, so manipulating its level is expected to affect only a fraction of PP1 pathways and therefore, it is not surprising that the effects are moderate.

Our finding that miR-183/96/182 overexpression improves memory differs from a previous report showing that miR-182 overexpression impairs long-term auditory fear memory and that its level is lower in the lateral amygdala after auditory fear conditioning[22]. The reason for this difference is unknown but may result from differential regulation of miR-183/96/182 or miR-182 in the cluster in different brain areas, and/or their action on different targets. While the study by Griggs et al.[22] employed amygdala-dependent auditory fear-conditioning paradigm, our study utilized an hippocampus-dependent object recognition task. Future studies examining the role of miR-183/96/182 cluster in other hippocampus-dependent and -independent learning paradigms should help determine the exact role of the cluster in different cognitive processes.

The present results also provide a link between the miRNA cluster and one of its targets—HDAC9—in memory regulation. Interfering with the interaction between miR-182 and HDAC9 alters object exploration, without affecting long-term memory. This is surprising, as miRNA modulation alone does not influence exploration behaviour in our paradigm. However, since object exploration is a prerequisite for proper learning and memory formation is preceded by learning and does depend on it, our results place HDAC9 as one of the early targets of miR-183/96/182 cluster involved in long-term memory. This in turn suggests that miR-183/96/182 cluster likely mediates its effects on memory formation through the concerted action of several target genes. These findings are relevant for cognitive functions beyond those assessed with our model, since both miR-182 and HDAC9 have also been implicated in schizophrenia. HDAC9 is one of a few genes with rare copy number variation in schizophrenia patients[49] and is hemizygously deleted in a small proportion of patients[32]. Further, disrupted hippocampal miR-182 signalling has been linked to changes in gene expression observed in schizophrenia and other mental illnesses[50]. A single-nucleotide polymorphism in this miRNA is predicted to be among key single-nucleotide polymorphisms linked to the disease[51]. Further studies on the interaction between miR-182 and HDAC9 may shed light on the regulation of cognition and cognitive disorders.

Besides HDAC9, our results revealed numerous plasticity-related genes as targets of the miR-183/96/182 cluster including Cacnb4, Gabra1, Grm5 and Gria1. Most of them are key mediators of synaptic plasticity processes such as LTP and LTD. Future studies examining the regulatory functions of miR-183/96/182 cluster in synaptic mechanisms may help further explain the behavioural phenotypes.

Finally, the present results have implications for the epigenetic regulation of memory, and specifically, the role of PP1 in this regulation. PP1 is known to modulate histone acetylation by directly associating with HDACs[9,52] or by cross-talk with some chromatin targets that affect HDACs[53]. Our results suggest another indirect and previously unknown control of HDACs by PP1 involving miRNAs. Interestingly, inhibiting HDACs induces substantial upregulation of miR-183/96/182 cluster in neuroblastoma cells[54], suggesting a possible feedback regulatory loop between HDACs and miRNAs.

## Methods

**Animals.** To inhibit nuclear PP1 in the mouse hippocampus, transgenic mice carrying the PP1 binding domain of the nuclear inhibitor of PP1, NIPP1 (NIPP1*) fused to EGFP (NIPP1*-EGFP6) and placed under a tetO promoter, were crossed with mice expressing a reverse tetracycline-controlled transactivator 2 under the control of a CaMKIIalpha promoter[9]. Conditional expression of NIPP1* in double transgenic mice was achieved by feeding the animals with a diet containing doxycycline (Pelodis) for eight days (6 mg g$^{-1}$ of wet mouse chow). The mice were group-housed (four mice per cage) under a reverse 12 h light/dark cycle (25 °C, 55% humidity), with food and water ad libitum. Behavioural experiments were carried out on reverse tetracycline-controlled transactivator 2/NIPP1*-EGFP6 adult

male and control littermates (3–5 months old) of C57Bl/6 mice background, during the animals' dark cycle. All experiments were conducted by experimenters blind to genotype. Experiments and animal maintenance were conducted in compliance with the Federation of Swiss Cantonal Veterinary Office and approved by Zürich Cantonal Veterinary Office (54/2012).

**Object recognition task.** Object recognition training was conducted in a rectangular arena (60 cm × 50 cm × 45 cm), with grey, opaque walls and translucent plexiglass bottom, under which an infrared light source was placed. It was located in a dedicated behavioural room illuminated by a dim light. Before NOR training, each animal was handled daily for 4 min on 4 consecutive days. Then an open field test was conducted by placing each mouse in an empty arena for 10 min and measuring the overall locomotion activity. An additional habituation to the empty arena was conducted one day later, before training started. For training (acquisition), three different unfamiliar objects were placed in the centre of the arena in a triangular arrangement. Each animal was allowed to explore the objects for five 5 min sessions spaced by a 5 min interval (strong protocol), or a single 10 min session (weak protocol). Object memory was tested in a 5 min session for which one of the familiar objects was replaced with a novel object. For the strong protocol, object memory was tested 24 h after training, and for the weak protocol, memory was tested 24 h (test 1) and 48 h (test 2) after training. The time that an animal spent exploring each object during testing was measured manually and with a video tracking system (ViewPoint Behaviour Technology) by an experimenter blind to group assignment. Object memory was expressed as the proportion of time spent exploring the novel object compared with the time spent exploring all objects (discrimination ratio). The discrimination ratio was normalized taking the average value from control animals as 100%. Throughout all experiments, movement of each animal inside the arena was tracked by an infrared camera connected to a tracking software (ViewPoint Behaviour Technology) in an adjacent room. For miRNA expression experiments, NOR controls (identified as 'habituation only' or 'non-trained' groups) were subjected to identical procedures but with no exposure to objects.

**Virus vector design and production.** For miRNA overexpression experiments, the scAAV2-EF1a-pri-miR-183/96/182-GFP construct was produced by cutting scAAV2-MCS (Cell Biolabs) by BalI/NotI (New England Biolabs). The transgene cassette containing Ef1a promoter (sequence from pEGP-mmu-miR-182 plasmid; Cell Biolabs), engineered truncated (T)-pri-miR-183/96/182, EGFP (from pEGP-mmu-miR-182 plasmid; Cell Biolabs), WPRE motif and 5′-BalI and 3′-NotI adaptors was chemically synthesized by GENEWIZ (South Plainfield, USA) and cloned into the scAAV backbone. Sequence of T-pri-miR-183/96/182 was as follows with mature miRNAs underlined:5′cctctgcagggtctgcaggctggagagtgtgactcctgtcctgt**tatggcactggtagaattcactg**tgaacagtctcagtcagtgaattaccgaagggccataaacagagcagagacagatccgcgagcaccttggagctcctcacccctttctgcctagacctctgtttccaggggtgccagggtacaaagacctcctctgctcctttcccccagagggcctgttccagtaccatctgcttggccgatt**tttggcactagcacattttgcttgtgtct**ctccgctgtgagcaatcatgtgtagtgccaatatgggaaaagcgggctgctgcgggccacgttcacctcccccggcatcccataataaaaaaacaagtatgctggaggcctcccaccatttttttggcaatggtagaactcacaccggtaaggtaatgggacccggtggttctagacttgccaactatggtgtaagtgctgagct. The scAAV2-EF1a-pri-miR-183/96/182-GFP allowed the generation of mature miR-183-5p, miR-96-5p and miR-182-5p sequences annotated in miRBASE v.20 (www.mirbase.org), whose expression was verified both by reverse transcription (RT)-qPCR (Supplementary Fig. 9). Control scAAV2-EF1a-control-GFP construct contained fragment of beta-globin intron (sequence from pEGP-mmu-miR-182 plasmid; Cell Biolabs) of length corresponding to T-pri-miR-183/96/182.

To suppress the level of endogenous miRNAs, a triple sponge for the miR-183/96/182 cluster containing four binding sites for each miRNA (perfectly complementary and containing a bulge) separated by a 15 nt spacer sequence were prepared as described previously[23]. The sponge sequences were assembled in pJ341 plasmids, excised at Hind III sites, and cloned into a p56 plasmid upstream of EGFP ORF. Sequence integrity was verified with sequencing.

Self-complementary AAV production was performed by triple transfection of HEK 293T cells using polyethylenimine with a plasmid bearing the target sequences between the internal terminal repeats of scAAV2, the AAV-helper plasmid encoding Rep2 and Cap for serotype 8 and the pHGTI-Adeno1 plasmid harbouring helper adenoviral genes (both kindly provided by C Cepko, Harvard Medical School, Boston, MA, USA). Vectors were purified using a discontinuous iodixanol gradient (Sigma, Optiprep). Encapsidated deoxyribonucleic acid (DNA) was quantified by TaqMan RT-PCR following denaturation of the AAV particles by Proteinase K, and titres were calculated as genome copies (GC) per ml.

**Stereotaxic surgery and intrahippocampal injections.** To overexpress or knockdown miRNAs, viral vectors were prepared as described above. To interfere with miR-182-HDAC9 binding, custom designed miRCURY LNA microRNA Power Target Site Blockers were obtained from Exiqon. The sequences are: miR-182/HDAC9 TSB: TTTGGCAAAAGTGCTA; negative control TSB: ACGTCTATACGCCCA. The oligonucleotides were stereotaxically injected in CA1 region of the hippocampus at a concentration of 1 µg µl$^{-1}$, in TurboFect in vivo transfection reagent (Dharmacon). For injection, animals were anaesthetized with 3%

isofluorane (Attane) and placed in a stereotaxic frame. Anaesthesia was maintained with 1.5% isofluorane in 100% oxygen for throughout surgery. Injection was carried out by lowering a glass pipette (Blaubrand, cut to a 20 µm inner diameter) filled with virus/oligonucleotides and attached to an injection pump (Stoelting) through a predrilled hole at the following coordinate targeting CA1 region of the hippocampus (from Bregma): $AP$: $-2.0$ mm, ML: $\pm 1.5$ mm and $DV$: $-1.6$ mm. A total of 1 µl ($10^9$ GC ml$^{-1}$) of virus vector or 1.5 µl of TSB oligos was injected into each hippocampus at a rate of 0.2 µl min$^{-1}$. The glass pipette was left in place for an additional 5 min, then carefully withdrawn and the wound was closed. The animals were allowed to recover for up to 2 weeks in their home cage before behavioural testing.

**Cell culture.** Mouse neuroblastoma (N2A) cells were obtained from American Type Cell Culture Collection and cultured in Dulbecco's Modified Eagle's Medium (DMEM—high glucose), supplemented with 10% (v/v) foetal bovine serum (Gibco) and 1% antibiotic–antimycotic (Gibco). These cells were chosen for their fast growth, easy maintenance and transfection, and potential for neuronal differentiation making them a convenient model for studying miRNA biogenesis. The cells were proven free of mycoplasma contamination through regular tests (MycoFluor mycoplasma detection kit). Before the start of experiments, the cells were passaged 1:4 and split every 3 days for at least five passages. On transfection day, 150,000–300,000 cells were plated in six-well plates. Transfection of a pool of siRNAs targeting PP1γ 3′UTR (ThermoFischer Scientific, sequences in Supplementary material) or negative control siRNA (All Star negative control, Qiagen) was carried out with HiPerfect transfection reagent (Qiagen). In pri-miRNA processing assays, inhibition of nuclear PP1 was achieved by over-expression of a plasmid containing NIPP1 construct (Origene). Then the cells were returned to the incubator and allowed to grow for 72 h before harvest or further treatment. In KCl treatment conditions, 50 mM of KCl was added to plated cells 1 h before harvest (unless indicated otherwise). In ActD (Tocris) treatment conditions, the cells were treated with 5 µg ml$^{-1}$ of the drug prepared in dimethylsulphoxide 1 h before KCl treatment or harvest. Rescue of PP1γ knockdown was performed by co-transfecting a plasmid containing PP1γ open reading frame (Origene) with siRNA against PP1γ 3′UTR. The cells were harvested by removing the medium, washing with ice-cold PBS three times and lysing with Tri-reagent (for RNA extraction) or radio immunoprecipitation buffer (RIPA) (for protein extraction). All experiments were conducted on at least three replicates from different passage numbers and repeated at least three times.

**Pri-miRNA processing and HDAC9 target validation assays.** Pri-miRNA processing assays were conducted as previously described[26]. The assay quantifies Drosha processing of pri-miRNA based on the decrease in luciferase activity, which is inversely proportional to the processing of pri-miRNA by Drosha. Briefly, fragments of pri-mir-182 and pri-mir-183, and the control pri-miR-10b, containing the hairpin and 100 bp flanking sequence were amplified from genomic DNA. PCR products were digested with the respective restriction enzymes and inserted at MCS in pmirGLO vector (Dual Reporter Luciferase Assay System, Promega) downstream to firefly luciferase reporter. Cropping of the hairpin stem-loop of inserts results in destabilization of the firefly reporter resulting in decrease in firefly luminescence. The unperturbed Renilla reporter produces stable luminescence, which serves as internal normalization control. Dual-luciferase reporters with pri-mir-182 and pri-mir-183 were transfected in N2a cells using cationic liposomes (Lipofectamine 2,000 reagent, Invitrogen). PP1 manipulations were performed by simultaneously transfecting N2a cells with NIPP1 over-expressing plasmid (Origene). The cells were lysed 48 h post transfection with passive lysis buffer (Promega) treatment at room temperature for 10 min. Ty lysates were then transferred to a 96 well plate, and luciferase activities of firefly and Renilla were read through luminometer GloMax 96 (Promega) equipped with dual injections dispersing LAR II (for firefly luciferase quantification) and Glomax (for renilla luciferase quantification) reagents sequentially.

For validation of HDAC9 targeting by miR-182, the same vector system and cloning strategy as described above was used (with the exception that HDAC9 3′UTR sequence containing miR-182 target site was amplified from genomic DNA using the following primers and inserted in the vector: hdac9_F1_NheI-GGCATAgctagcAGGATATGTGCCAGGCAGTC, hdac9_R1_SaII-CGCTTAgtcgacAATGGGCGTCATTGTTCTTC). Dual-reporter vectors with 3′UTR HDAC9 inserts were transfected in N2a cells using cationic liposomes (Lipfectamine 2,000 reagent, Invitrogen). MiR-182 mimic (Qiagen) was simultaneously transfected to the cells. The cells were lysed 24 h post transfection with passive lysis buffer (Promega) treatment at room temperature for 10 min. The lysates were then transferred to a 96-well plate, and luciferase activity of firefly and Renilla were read through luminometer GloMax 96 (Promega) equipped with dual injections dispersing LAR II (for firefly luciferase quantification) and Glomax (for renilla luciferase quantification) reagents sequentially.

**RNA extraction and RT-qPCR.** Mouse hippocampal tissue was homogenized using TissueLyser (Qiagen) in Trizol reagent (Invitrogen). Total RNA was extracted by phenol–chloroform precipitation. For extraction from cells, the medium was removed, the cells were washed three times with ice cold-PBS, lysed

and homogenized by adding Trizol to the plates. Subcellular fractionation of nuclear and cytoplasmic RNA was performed using Norgen's Cytoplasmic and nuclear RNA purification kit (Norgen BioTek, Canada). One microgram of total RNA was treated with RNase free DNase (Promega) and reverse transcribed using miScript II RT kit (Qiagen). Miscript primer assays for mature and pre-miRs were used to amplify the respective transcripts from a complementary DNA (cDNA) pool (Supplementary Table 1). For mRNA quantification, custom designed gene specific primers were used (Supplementary Table 1). Real time PCR was performed on LightCycler 480 (Roche). The following pool of endogenous controls were used to normalize qPCR data as appropriate: RNU6, GAPDH, Tubd1 and 18S rRNA.

**Deep sequencing.** The quality and quantity of RNA was determined using a Bioanalyzer (Agilent) and Qubit fluorometer (Invitrogen) respectively. Small RNA libraries were prepared from 1 µg of total RNA using TruSeq Small RNA Kit (Illumina) according to manufacturer's instructions. Briefly, 3′ and 5′ adaptors were ligated to small RNAs using T4 DNA ligase. Ligated RNA was reverse transcribed to cDNA using superscript II reverse transcriptase. The resulting cDNA template was amplified by PCR to generate a cDNA library. For each sample, a common forward primer (which binds to the 5′ adaptor complement) and a unique reverse primer (which binds to the 3′ adaptor complement and contains a unique sequence for each sample) were used. The quality of the resulting amplicons was analysed on a high sensitivity DNA ChIP (Bioanalyzer). Next, the cDNA construct was run on a 6% polyacrylamide gel electrophoresis (PAGE) gel and amplicons corresponding to adaptor ligated miRNA sequences (145–160 nt) were excised out. DNA from excised gel was eluted and precipitated in ethanol. The size, purity and amount of cDNA was assessed on high sensitivity DNA ChIP (Bioanalzyer). Then a titration run was done to check quality of the library and validate the amount. Finally, multiplexed samples were sequenced on Hi-Seq 2,000 using TrueSeq SBM v5 sequencing kit. The resulting sequences were demultiplexed and sorted to individual samples according to their index codes. The error rate for each library was estimated based on a PhiX reference spiked before sequencing. Next, adaptors were trimmed from the reads and the resulting inserts were categorized by size. Bioinformatic analysis of sequencing reads was performed using ncPRO-seq pipeline[55]. After confirming that the majority of reads are attributed to the size range of miRNAs, the sequences were mapped and aligned to the mouse reference genome (NCBI37/mm9) using bowtie algorithm. The number of reads uniquely mapping to miRNAs were normalized to the total number of reads in each sample. Finally, differential expression analysis of miRNAs regulated upon expression of NIPP1* and/or during memory formation was performed using Wilcoxon unpaired test[56].

**Protein extraction and Western blot.** Total protein was extracted from N2A cells using RIPA with 1:1,000 protease inhibitor cocktail (Sigma-Aldrich) and 1:500 phenylmethyl sulphonyl fluoride. Cells were lysed directly on the culture plate with 100–150 µl RIPA and scraped off with a cell scraper. The lysate was transferred to a micro-centrifuge tube and sonicated for five cycles, each comprising 30 s of sonication with 30 s intervals. The resulting mixture was centrifuged for 15 min at 14,000g to separate the protein mixture (supernatant) from cellular debris (pellet). Proteins (20–40 µg) were resolved on SDS–PAGE and transferred onto a nitrocellulose membrane (Bio-rad). Membranes were blocked in 3% BSA for 1 h, and then incubated in primary (overnight at 4 °C) and secondary (1 h at room temperature) antibodies. They were scanned using Odessey IR scanner (Li-Cor Bioscience), and band intensity was determined and quantified using image analysis software (ImageJ). The following antibodies were used: primary-HDAC9 (Abcam, ab59718), PP1γ (Millipore, 07-1218), GAPDH (Abcam, ab9485), beta actin (Abcam, ab8226); secondary-anti-mouse IRDye goat anti-mouse (LI-COR, 925-32210) and IRDye goat anti-rabbit (LI-COR, 925-32211). Original blots are shown in (Supplementary Figs 15 and 16).

**Statistical analyses.** For deep sequencing data, differential expression between two groups was assessed by Wilcoxon signed-rank test. For qPCRs, Western blotting and behavioural experiments with two groups, or with more than two groups in which *a priori* only one group was expected to give a difference (that is, control, knockdown, rescue of knockdown like in Fig. 2c), two-tailed Student *t* test was used. Analysis of variances (ANOVAs) followed by Fisher's least significant difference were also conducted. For other experiments, one- or two-way ANOVAs were used followed by protected Fisher's least significant difference and Tukey's *posthoc* analyses when appropriate (see Supplementary Table 2). Outliers were defined as values beyond two s.d. from a group mean and were removed from the analyses. Significance was set at $P \leq 0.05$ for all tests. Statistical analyses were performed using GraphPad prism or *R* statistical software.

**Data availability.** All next generation sequencing data has been deposited in the Gene Expression Omnibus (GEO) and can be accessed using GEO accession number GSE83707. All relevant data from this study are available from the corresponding author upon request.

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

## Acknowledgements

This work was supported by the University of Zurich, the Swiss Federal Institute of Technology, the Swiss National Science Foundation and the National Competence Center for Research 'Neural plasticity and Repair'. We thank Yannick Rotharcher, Cécile Hauser, Jennifer Brown and Lubka Spassova for technical help, Jean-Claude Paterna for virus production, Gregor Fisher and animal caretakers for mouse colony management, Witold Filipowicz and Olivier Voinnet for constructive discussion and critical reading of the manuscript.

## Author contributions

B.T.W. and I.M.M. initiated and designed the study, analysed the results and wrote the manuscript. B.T.W. performed behavioural experiments and stereotaxic injections, sequencing and molecular analyses. A.J. designed and executed *in vitro* experiments, and helped in the interpretation and drafting of *in vitro* data. N.G. and A.J. conducted RT-PCR experiments for *in vivo* miRNA target validation and miRNA expression after weak memory protocol respectively. E.A.K. designed cloning experiments and conducted them with A.J. J.K. produced virus vectors and constructs for miRNA overexpression and sponge. A.M. analysed sequencing data.

## Additional information

**Competing financial interests:** The authors declare no competing financial interests.

