## [Peer Review File · Nature Communications]

Editorial Note: this manuscript has been previously reviewed at another journal that is not operating a transparent peer review scheme. This document only contains reviewer comments and rebuttal letters for versions considered at *Nature Communications*.

Reviewer comments:

Reviewer#1

The authors have addressed some - but not all - of my concerns. The preliminary nature of the report has been addressed. In particular, the addition of the new "sponge" data significantly strengthens the manuscript by establishing that constitutively expressed miR-183/96/182 plays a role in new learning/memory, rather than simple ectopic expression impacting processes that otherwise they would play no role in. Similarly, the new data supporting in vivo targeting of HDAC9 also strengthens the manuscript. Some assessment of alterations in synaptic processing would have represented an important addition to the paper, despite the authors' protestations otherwise, but it is hoped that synaptic mechanisms that explain the action of the miRs will represent the subject of a thorough future report.

Reviewer #2

The authors have been responsive to earlier concerns however, important concerns remain.

1. The miR-182 targeting of HDAC9 study.

The authors say: Interfering with miR-182 targeting of HDAC9 in mice overexpressing miR-183/96/182 cluster significantly reduced object exploration behavior during training and testing, without affecting novel object discrimination (Fig. 7 c-e). Together, these results identify HDAC9 as one of the mediating factors of miR-182/96/183 on cognitive processes.

Throughout the paper the authors use discrimination as an index of memory (e.g. Fig. 5). Suddenly, object exploration is used as a measure of cognition and the fact that discrimination is unaffected is completely ignored.

2. The authors still do not provide justification for using Fisher vs Tukey post-hoc tests. If the analysis is the same and LSD tests are no more recommended, it is not clear why the authors still present the LSD instead of the Tukey results. Also, while post-hoc analyses are indeed tolerated when overall ANOVA is insignificant, LSD is one exception where this is not allowed because the first step of the protected LSD test is to check if the overall ANOVA rejects the null hypothesis of identical means. If it doesn't, individual comparisons should not be made.

3. I still find the behavioral phenotype very weak and transient. Demonstrating the significance of the suggested pathway in another form of hippocampus-dependent memory would have significantly strengthened the impact of this work.

Reviewer#3

This revised manuscript reports that memory control by PP1 involves the microRNA cluster miR-183/96/182 that is regulated during memory formation. Inhibiting nuclear PP1 in neurons in adult mice in vivo or training wild-type animals on an object recognition task similarly increases the level of pre-miR-183/96/182 in the hippocampus. Mimicking this increase by overexpressing miR-183/96/182 in the hippocampus enhanced object memory. Further, PP1 controls these miRNAs through transcription-independent means involving the processing of these miRNAs. They propose that miRNAs play a role in memory formation and suggest the implication of PP1 in miRNAs processing in the adult brain.

This paper has been appropriately revised.

Comments from reviewer#1 in response to issues raised by reviewer#2

My own evaluation of the manuscript, and my ultimate recommendation, was based on the fact that the reviewers sought to address two of my major concerns:

1. They analyzed the effects of loss-of-function of the miR-183/96/182 cluster on novel object recognition and generated positive data.
2. They generated data to better link miR-183/96/182 to HDAC9 regulation and a role for HDAC9 in the effects of the miR cluster.

As these data alleviated my major concerns, I recommended that the manuscript be accepted for publication.

However, even with the addition of these new data the other reviewer still had a number of concerns:

1. The new data supporting a link between miR-182 and HDAC9 (biochemical and behavioral).

In response to this concern the authors conducted a new experiment to demonstrate that miR-182 acts through HDAC9.

The other reviewer is not satisfied that these new data are sufficiently robust to support the authors' conclusions.

I can understand the reviewer's concern in this regard.

The discrimination index, which was used as a measure of learning/memory throughout, was not affected.

However, it is very difficult to determine precisely how miRNAs impact behavior as their targeting of protein-coding transcripts is so pleiotropic.

It is a tall order to ask the authors to fully characterize the mechanisms by which miR-182 acts, as it may target multiple transcripts simultaneously to modify behavior. Hence, establishing some role for HDAC9 in the actions of miR-182 is, in my view, sufficient for publication.

As such, I do not consider the concern expressed here by the reviewer to be sufficient to block publication.

2. The concern related to statistical analyses is warranted; the reviewer makes a fair point. The authors should analyze their data based on the recommendation of the reviewer. Also, they should have addressed this concern in their rebuttal letter.

I believe that this point needs to be properly addressed before the manuscript can be considered ready for publication.

3. Related to the above point, and the reviewer's third concern, the effect sizes are, indeed, relatively modest.

In my view, this is the most pressing concern but the most challenging to address in review.

Given the modest magnitude of effect, and the fact that data are analyzed as a % value, one wonders if the data would be reproducible in other laboratories.

Hence, I do agree with the reviewer here; demonstrating that some other form of hippocampal-dependent learning is impacted would increase confidence in the findings.

Therefore, on balance, I can understand entirely the reviewer's concerns and consider the third point the most important.

Response to Reviewers' comments

Reviewer #1

Comment: The authors have addressed some - but not all - of my concerns. The preliminary nature of the report has been addressed. In particular, the addition of the new "sponge" data significantly strengthens the manuscript by establishing that constitutively expressed miR-183/96/182 plays a role in new learning/memory, rather than simple ectopic expression impacting processes that otherwise they would play no role in. Similarly, the new data supporting in vivo targeting of HDAC9 also strengthens the manuscript. Some assessment of alterations in synaptic processing would have represented an important addition to the paper, despite the authors' protestations otherwise, but it is hoped that synaptic mechanisms that explain the action of the miRs will represent the subject of a thorough future report.

Response: The reviewer is right that assessing alterations in synaptic processing would be a useful addition to the paper. It would however be a whole study on its own that would address another question than the one studied in this manuscript and is therefore outside the scope of this manuscript. We also hope to be able to conduct such analyses on synaptic mechanisms in the near future.

Reviewer #2

Comment: The authors have been responsive to earlier concerns however, important concerns remain. The miR-182 targeting of HDAC9 study. The authors say: Interfering with miR-182 targeting of HDAC9 in mice overexpressing miR-183/96/182 cluster significantly reduced object exploration behavior during training and testing, without affecting novel object discrimination (Fig. 7 c-e). Together, these results identify HDAC9 as one of the mediating factors of miR-182/96/183 on cognitive processes. Throughout the paper the authors use discrimination as index of memory (e.g. Fig. 5). Suddenly, object exploration is used as a measure of cognition and the fact that discrimination is unaffected is completely ignored.

Response: Both the discrimination ratio and object exploration are actually examined in all behavioral experiments using the novel object recognition test, including those examining the effects of miR-182/HDAC9 target site blockade in mice overexpressing miR-183/96/182 (Fig. 7c-e), we are sorry if the reviewer thought we changed measures along the manuscript. The results of the experiments are however different depending on the gene that is manipulated. While miR-183/96/182 overexpression or knock-down affects the discrimination ratio indicating an effect on object memory, miR-182/HDAC9 target site blockade in mice overexpressing miR-183/96/182 affects object exploration itself. The fact that the discrimination ratio is not changed in this case indicates that memory formation itself is not impaired but object learning is. This is mentioned in the results section and the discussion of the manuscript (page 11 and 14).

Comment: The authors still do not provide justification for using Fisher vs Tukey post-hoc tests. If the analysis is the same and LSD tests are no more recommended, it is not clear why

the authors still present the LSD instead the Tukey results. Also, while post-hoc analyses are indeed tolerated when overall ANOVA is insignificant, LSD is one exception where this is not allowed because the first step of the protected LSD test is to check if the overall ANOVA rejects the null hypothesis of identical means. If it doesn't, individual comparisons should not be made.

Response: The initial choice of Fisher's LSD posthoc tests was motivated by the study design and the type of statistical analyses that were required but we agree with the reviewer that in some cases, Tukey's post-hoc analyses can also be conducted, which we now did as requested. The reviewer is also right that post-hoc analyses are tolerated when overall ANOVA is insignificant, which is indeed the case for unprotected Fisher's LSD used in some of the analyses in the manuscript, but that protected Fisher's LSD shall be used only when ANOVAs are significant. In the revised manuscript, we now use Tukey's post-hoc tests as recommended by the reviewer in all cases requiring multiple comparisons between groups when the ANOVA is significant. These analyses were done on data in Figure 2d, e, Supplementary Figure 7 and Supplementary Figure 8a (right panel). In all these cases, Tukey's post-hoc analyses confirmed the significance of the effects obtained with protected Fisher's LSD analyses done previously. In cases not requiring multiple comparisons between groups because the study design expected *a priori* only one group to be different from the others, Fisher's LSD analyses were chosen over Tukey's post-hoc analyses because they are preferred for such study design. To validate statistical significance in all these cases, classical t-tests were conducted and did confirm the significance observed with Fisher's LSD done previously (Figures 2c and 6c). The results of re-assessed statistical analyses are reported in new Supplementary Table 2 where t-tests, Fisher's LSD and Tukey's post-hoc analyses are indicated for each relevant dataset. Data from figures not indicated in the table did not require re-analysis because they did not involve ANOVAs and were not questioned by the reviewer. The description of statistical analyses has also been updated in the methods section.

Comment: I still find the behavioral phenotype very weak and transient. Demonstrating the significance of the suggested pathway in another form of hippocampus-dependent memory would have significantly strengthened the impact of this work.

Response: The reviewer states that the behavioral phenotype is very weak and transient but the behavioral data reported in the manuscript show robust and persistent effects on memory so we believe that this may be a mis-reading of the data. Thus, both the overexpression and the knock-down of miR-183/96/182 cluster are shown to affect object memory performance, with respectively, an improvement and an impairment of almost 20% and for respectively, 48h and 24h (Figure 5 and Supplementary Figure 11 and 12). Likewise, miR-182/HDAC9 blockade impairs performance by about 20% after 24h (Figure 7). On an object recognition task, 20% change in performance is substantial, in particular for an improvement in memory because the task is subtle and driven purely by cognitive processes (unlike other tasks like fear conditioning which has a strong emotional and physical component). The size and duration of these effects are remarkable and indicate a solid and long-lasting behavioral phenotype. Further, we regret that the reviewer did not take into account the demonstration of the bidirectionality of the effect of miR-183/96/182 overexpression or knock-down on memory performance (improvement in one case, impairment in the other). Such clearly opposite effect on memory is a very strong indication that it is due to the cluster itself, and is not an unspecific

effect of the manipulation, which is an extremely solid demonstration. Such demonstration is rare in the field and is therefore of special value in this manuscript. Further, we acknowledge the suggestion of the reviewer to add more behavioral data and agree that it would be nice to have data on another hippocampus-dependent task to duplicate and confirm the findings. However, we strongly believe that such data are not vital for the paper, and would only bring a small increment to the present behavioral results, which are already extensive. They would also require a disproportionately large amount of work, mice and money. The present revised version indeed already contains additional behavioral data that the reviewer asked for in a previous revision, in particular, behavioral testing with 2 different protocols and with a time course (Supplementary Figure 8 and 13). These additional data did confirm the memory effect so we hope that the reviewer will be satisfied by the present behavioral data.

Reviewer#3

No more comment from this reviewer.

Comments from reviewer#1 in response to issues raised by reviewer#2

Comment: My own evaluation of the manuscript, and my ultimate recommendation, was based on the fact that the reviewers sought to address two of my major concerns:

- 1. They analyzed the effects of loss-of-function of the miR-183/96/182 cluster on novel object recognition and generated positive data.*
- 2. They generated data to better link miR-183/96/182 to HDAC9 regulation and a role for HDAC9 in the effects of the miR cluster.*

As these data alleviated my major concerns, I recommended that the manuscript be accepted for publication. However, even with the addition of these new data the other reviewer still had a number of concerns:

- 1. The new data supporting a link between miR-182 and HDAC9 (biochemical and behavioral). In response to this concern the authors conducted a new experiment to demonstrate that miR-182 acts through HDAC9. The other reviewer is not satisfied that these new data are sufficiently robust to support the authors' conclusions. I can understand the reviewer's concern in this regard. The discrimination index, which was used as a measure of learning/memory throughout, was not affected. However, it is very difficult to determine precisely how miRNAs impact behavior as their targeting of protein-coding transcripts is so pleiotropic. It is a tall order to ask the authors to fully characterize the mechanisms by which miR-182 acts, as it may target multiple transcripts simultaneously to modify behavior. Hence, establishing some role for HDAC9 in the actions of miR-182 is, in my view, sufficient for publication. As such, I do not consider the concern expressed here by the reviewer to be sufficient to block publication.*

Response: We thank Reviewer 1 for considering that the demonstration that HDAC9 has a role in the action of miR-182 is sufficient for publication and that the concern expressed by Reviewer 2 should not prevent publication.

Comment: 2. The concern related to statistical analyses is warranted; the reviewer makes a

fair point. The authors should analyze their data based on the recommendation of the reviewer. Also, they should have addressed this concern in their rebuttal letter. I believe that this point needs to be properly addressed before the manuscript can be considered ready for publication.

Response: As requested, statistical analyses were re-done with Tukey's post-hoc analyses based on the recommendation by Reviewer 2.

Comment: 3. Related to the above point, and the reviewer's third concern, the effect sizes are, indeed, relatively modest. In my view, this is the most pressing concern but the most challenging to address in review. Given the modest magnitude of effect, and the fact that data are analyzed as a % value, one wonders if the data would be reproducible in other laboratories. Hence, I do agree with the reviewer here; demonstrating that some other form of hippocampal-dependent learning is impacted would increase confidence in the findings. Therefore, on balance, I can understand entirely the reviewer's concerns and consider the third point the most important.

Response: We understand that the reviewer may consider the effect size to be modest. However when examined closely and considered altogether, the behavioral effects are actually solid and persistent. The results show that both the overexpression and the knock-down of miR-183/96/182 cluster affect object memory performance, with respectively an improvement and an impairment of almost 20% for respectively, 48h and 24h which is a long-term effect (Figure 5 and Supplementary Figure 11 and 12). Further, miR-182/HDAC9 blockade also impairs performance by about 20% for 24h (Figure 7). On an object recognition task, 20% change in performance is substantial, in particular for an improvement in memory because the task is subtle and driven purely by cognitive processes (unlike other tasks like fear conditioning which has a strong emotional and physical component). The size and duration of these effects are therefore significant and indicate a robust and long-lasting behavioral phenotype. Further, the fact that miR-183/96/182 overexpression improves performance while its knock-down has an opposite effect and impairs performance is a very strong indication that the effect is real and directly due to the level of miR-183/96/182. This again indicates that the effect on memory is strong. For all these data, percent of novel object discrimination index is used since this is a well accepted and intuitive measure of memory for the NOR paradigm. We agree that demonstrating that another form of hippocampus-dependent learning is impacted would increase confidence in the findings. However, this would clearly require a disproportionately large effort, time and money for only a small increment to the present behavioral results. This would require a huge amount of work including surgically-cannulating mice for injection with 6 different viral vectors (miR mimics, miR sponge, miR-182/HDAC9, and 3 respective controls), checking for virus expression, testing behavior on 10-14 mice for each vector, and would take up to 1 year and use many animals. We would like to stress that this study already contains an unusually massive amount of data including transcriptomic, molecular, biochemical and behavioral results, which required 2 transgenic mouse lines, 6 different virus-injected mouse lines (almost 500 mice used in total), 4 different cell transfection vectors, and 51 different primers, and considerable intellectual and technical effort. We believe therefore that adding more behavior would not significantly strengthen the paper and hope that the reviewer will agree with us.

REVIEWERS' COMMENTS:

Reviewer #2 (Remarks to the Author):

In their rebuttal, the authors have argued that the evidence that they present is sufficiently compelling that the additional experiment requested by both reviewers - showing a role of the cluster in another hippocampal-dependent behavior - is not required. As this experiment was not executed, the same issues raised on the first revised manuscript remain.

Reviewer #3 (Remarks to the Author):

The authors have addressed my key concern related to the statistical analyses. The other issues have been addressed in the results/discussion. I have no remaining concerns.

Response to Reviewers' remarks

Reviewer #2

Remark: In their rebuttal, the authors have argued that the evidence that they present is sufficiently compelling that the additional experiment requested by both reviewers - showing a role of the cluster in another hippocampal-dependent behavior - is not required. As this experiment was not executed, the same issues raised on the first revised manuscript remain.

Response:

We regret that the reviewer is still not satisfied with the current behavioral data, and requests more data. As addressed in our previous response, we believe that the current data included in this manuscript are sufficient to support the conclusions that PP1 acts on memory through miR-183/96/182. Additional data would not make the conclusions stronger and considering the huge effort that would require to collect them (several viral constructs to inject, many animals, about 1 year of work), we feel that they are not justified.

Reviewer #3

Remark: The authors have addressed my key concern related to the statistical analyses. The other issues have been addressed in the results/discussion. I have no remaining concerns.

Response: We thank the reviewer for acknowledging our revisions.